# UMVMap: Improving Vectorized Map Construction via Multi-vehicle Perspectives

## Abstract

Prevalent vectorized map construction pipelines predominantly follow an end-to-end DETR-based paradigm. While these methods have achieved significant advancements, they are limited by their reliance on data from a single ego vehicle, which restricts their effectiveness and can lead to perceptual uncertainty in handling complex environmental scenarios. To address this limitation, we introduce a novel framework: Uncertainty-aware Multi-Vehicle Vectorized Map Construction (UMVMap). This framework effectively mitigates uncertainties by leveraging relevant non-ego information. UMVMap comprises two essential components: the Uncertainty-aware Multi-Vehicle Vectorized Map Construction Network (UMVMap-Net), which optimally integrates data from multiple vehicles, and the Uncertainty-aware Non-ego Vehicle Selection (UNVS) strategy, which identifies and incorporates the most informative non-ego data to minimize uncertainty. Comprehensive evaluations on the nuScenes dataset demonstrate that UMVMap significantly outperforms the single-vehicle MapTRv2 baseline by a margin of 9.1% and 9.9% respectively on the full and partial validation sets, with each of its components proving to be both effective and robust.

## 1 Introduction

Vectorized maps are crucial for autonomous driving Antonello et al. (2017); Gao et al. (2020), providing abundant road details and rich semantic information that support tasks such as self-localization Levinson et al. (2007) and path planning Da & Zhang (2022). Traditionally, vectorized maps are generated using simultaneous localization and mapping (SLAM)-based techniques Shan & Englot (2018); Shan et al. (2020), which require subsequent refinement by human operators and incur high maintenance costs. In recent years, numerous approaches Liao et al. (2022); Ding et al. (2023); Zhang et al. (2024b) have adopted end-to-end learning-based methods for vectorized map construction, often utilizing a DETR-based architecture Carion et al. (2020).

Despite their advancements, these methods are constrained by relying solely on the perspective of a single ego vehicle, which limits their ability to handle complex environmental conditions. As illustrated in Fig. 1 (a), in scenarios involving occlusion, single-vehicle approaches, such as MapTRv2 Liao et al. (2023), tend to produce ambiguous perceptions in the occluded areas, as these regions are beyond the visibility of the ego vehicle. Additionally, the map element perception of single-vehicle models becomes increasingly uncertain as the distance from the ego vehicle grows, often leading to the omission of distant map elements in the constructed map. To illustrate this, we conducted a quantitative analysis by evaluating the performance of the single-vehicle method across a range of perception distances. As shown in Fig. 2 (a), the single-vehicle method exhibits a noticeable decline in performance as the perception range increases. In contrast, the multi-vehicle paradigm demonstrates a much smoother performance curve, suggesting that integrating auxiliary information from non-ego perspectives can effectively mitigate the uncertainties inherent in the ego vehicle's perspective. Also, it is intuitive that by observing from varied non-ego perspectives, the uncertainty of the occluded areas can be effectively reduced. Furthermore, Fig. 2 (b) presents histograms depicting the number of ego vehicles with varied non-ego vehicle numbers at various distance ranges. It is evident that across all distance intervals, the majority of ego vehicles are able to obtain support from more than 10 non-ego vehicles, which underscores the feasibility of leveraging information from non-ego vehicles as a supplementary aid.

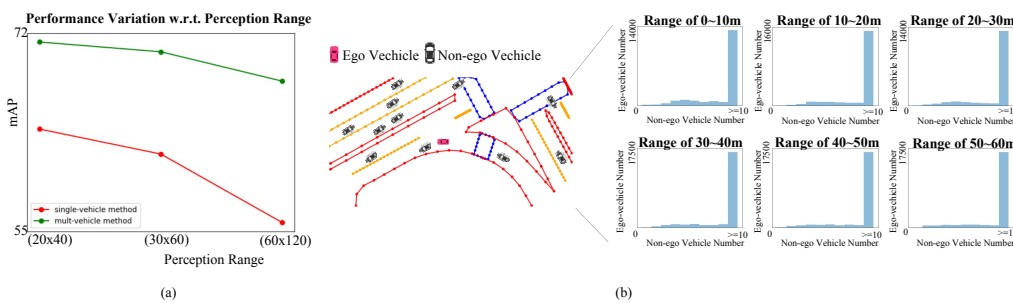

Figure 1: (a) Single-vehicle approaches, such as MapTRv2, tend to produce ambiguous perceptions in the occluded or distant areas. (b) In contrast, our UMVMap is capable of addressing uncertain details with the aid of proper non-ego information.

Figure 2: (a) The performance variation with respect to the perception range of a single-vehicle method and a multi-vehicle one. (b) Statistical analysis indicates that most ego vehicles can receive assistance from over 10 non-ego vehicles at various distances, which validates the feasibility of leveraging non-ego information as a supportive measure. Each histogram represents the statistical outcomes for different perceptual range intervals. In these histograms, the horizontal axis denotes the number of available non-ego vehicles, while the vertical axis represents the number of ego vehicles corresponding to each count of available non-ego vehicles.

To this end, we propose a novel Uncertainty-aware Multi-Vehicle vectorized Map Construction (UMVMap) framework, which is capable of addressing uncertain details with the aid of proper non-ego information. The UMVMap consists of two key components: the Uncertainty-aware Multi-Vehicle Vectorized Map Construction Network (UMVMap-Net), which optimally utilizes multi-vehicle data, and the Uncertainty-aware Non-ego Vehicle Selection (UNVS) strategy, which identifies the most beneficial non-ego data to reduce uncertainty. Specifically, UMVMap-Net is composed of an Uncertainty-guided Multi-Vehicle Information Integrator (UMVII) and a Segmentation Prior Guided Map Decoder (SPMDec). UMVII first integrates multi-vehicle data, while SPMDec decodes this integrated information to produce the vectorized map. Additionally, the UNVS strategy is designed to select non-ego vehicles that provide complementary information, particularly in areas where the ego vehicle's perception is highly uncertain.

The proposed UMVMap is evaluated on the large-scale nuScenes Caesar et al. (2020) datasets. Experimental results demonstrate that UMVMap significantly outperforms the single-vehicle MapTRv2 baseline by a margin of 9.1% and 9.9% respectively on the full and partial validation sets. In addition, a comprehensive ablation analysis is carried out and the promising results prove the effectiveness and robustness of our proposed key components within UMVMap.

The main contributions of this study can be summarized as follows:

- An Uncertainty-aware Multi-Vehicle vectorized Map Construction (UMVMap) framework is proposed to address uncertain details by leveraging non-ego information. To the best of our knowledge, UMVMap is the first framework to handle vectorized map construction using multi-vehicle data.

- A Uncertainty-aware Non-ego Vehicle Selection (UNVS) strategy is introduced to select non-ego vehicles that provide complementary information, especially in regions where the ego vehicle's perception is highly uncertain.

- Comprehensive evaluations on the nuScenes datasets demonstrate that UMVMap significantly outperforms the single-vehicle MapTRv2 baseline by a margin of 9.1% and 9.9%

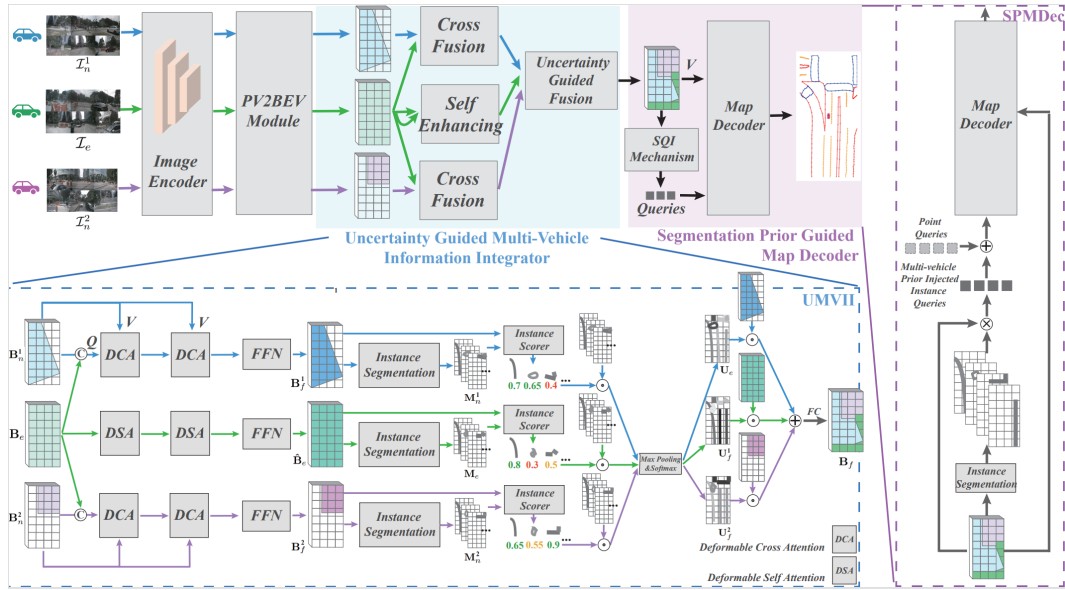

Figure 3: **Overview of UMVMap-Net.** UMVMap-Net comprises an Uncertainty-guided Multi-Vehicle Information Integrator (UMVII), which properly integrates information from multi-vehicle data, and a Segmentation Prior Guided Map Decoder (SPMDec), which decodes the integrated information from UMVII to construct the vectorized map.

respectively on the full and partial validation sets, with each component proving to be both effective and robust.

## 2 RELATED WORKS

**Vectorized Map Construction.** Vectorized maps are conventionally generated using simultaneous localization and mapping (SLAM)-based techniques Shan & Englot (2018); Shan et al. (2020), and then refined through human intervention, which suffers from high maintenance expenses. In recent years, there has been a surge of interest in utilizing onboard sensors for vectorized map construction. HDMapNet Li et al. (2022) generates Bird's-Eye-View (BEV) semantic segmentations and then performs a heuristic, time-consuming post-processing step to produce vectorized map instances. Beyond this, VectorMapNet Liu et al. (2023) introduces the first end-to-end vectorized map construction framework, employing auto-regression to sequentially predict points. Subsequently, an increasing number of recent approaches Liao et al. (2022); Ding et al. (2023); Liao et al. (2023); Zhang et al. (2024b) have adopted end-to-end methods for vectorized map construction, with many leveraging a DETR-based architecture Carion et al. (2020). However, these methods are constrained by their reliance on a single input and struggle to manage challenging environmental conditions, such as occlusions or adverse weather. StreamMapNet Yuan et al. (2024) takes a significant stride forward by incorporating temporal information through a streaming strategy, aiming to enhance the temporal consistency and quality of vectorized maps. Nevertheless, in some hard cases, constructing a high-quality vectorized map from the perspective of a single vehicle remains a formidable task. To this end, in this paper, we propose to mitigate uncertainties by leveraging relevant non-ego information, and collaboratively constructing superior vectorized maps.

**Collaborative Perception for Autonomous Driving.** Collaborative perception Han et al. (2023) in intelligent transportation systems, which integrates information from multiple vehicles to enhance environmental understanding, has been researched for over a decade despite being limited by the lack of effective techniques and large-scale datasets Rauch et al. (2012); Kim et al. (2014). Existing collaborative perception methods, based on the message-sharing strategy, can be broadly classified into early (data-level) Chen et al. (2019), intermediate (feature-level) Xu et al. (2022); Yang et al. (2023) and late (object-level) Xu et al. (2023) ones. Among these, intermediate collaboration is

preferred for its optimal balance between performance and transmission bandwidth. In this case, the concept of collaborative perception has been adopted in various tasks related to autonomous driving. CoCa3D Hu et al. (2023) enhances camera detection capabilities through multi-agent collaboration, leading to more comprehensive 3D detection. CoHFF Song et al. (2024) enhances camera-based semantic occupancy prediction through collaborative perception, achieving greater accuracy and comprehensiveness. This improvement is driven by the hybrid fusion of semantic and occupancy features, along with the shared compressed orthogonal attention features between vehicles. By constructing collaborative neural rendering field representations to recover failed perceptual messages sent by multiple agents, RCDN Wang et al. (2024) achieves a robust camera-insensitivity collaborative perception for 3D Neural modeling. Aside from prior studies, we are pioneering the exploration of vehicle-to-vehicle collaborative perception in vectorized map construction.

## 3 METHODOLOGY

### 3.1 PROBLEM SETUP

Given surround-view images $\mathcal{I}_e = \{\mathbf{I}_e^i\}_{i=1}^K$ captured from onboard cameras on an ego vehicle along with surround-view images $\{\mathcal{I}_n^j\}_{j=1}^{N_n} = \{\{\mathbf{I}_n^{j,i}\}_{i=1}^K\}_{j=1}^{N_n}$ captured from $N_n$ non-ego vehicles, the goal of collaborative vectorized map construction is to detect local BEV map instances while locating their corresponding structures, where $K$ is the number of views and $N_n$ is the number of non-ego vehicles. The final output of the model is a set of detected map elements. Each map element in the output set consists of a class label $c$ and an ordered sequence of points $\mathbf{P} = \{(x_i, y_i)\}_{i=1}^{N_{pt}}$ that represents its structure, with $N_{pt}$ denoting the number of points for each map element.

### 3.2 FRAMEWORK OVERVIEW

The overall UMVMap framework includes an Uncertainty-aware Multi-Vehicle vectorized Map Construction Network (UMVMap-Net) and an Uncertainty-aware Non-ego Vehicle Selection Strategy. UMVMap-Net is depicted in Fig. 3, which is primarily composed of an Uncertainty-guided Multi-Vehicle Information Integrator (UMVII) and a Segmentation Prior Guided Map Decoder (SP-MDec). First, UMVII is employed to properly integrate information from multi-vehicle data. Subsequently, SPMDec decodes the integrated information from UMVII to construct the vectorized map. Furthermore, aside from properly integrating multi-vehicle information, selecting non-ego vehicles that provide complementary information in regions where the ego vehicle's perception is highly uncertain can also significantly enhance the quality of the vectorized map construction. To achieve this, we further design the Uncertainty-aware Non-ego Vehicle Selection Strategy (illustrated in Fig. 4).

### 3.3 UNCERTAINTY-AWARE MULTI-VEHICLE VECTORIZED MAP CONSTRUCTION NETWORK

**BEV Encoder.** Initially, UMVMap-Net extracts BEV features from both ego and non-ego data. To ensure a fair comparison, we adopt the BEV feature extraction paradigm as implemented in MapTRv2 Liao et al. (2023). Given the ego and non-ego multi-view images $\mathcal{I}_e$ and $\{\mathcal{I}_n^j\}_{j=1}^{N_n}$, multi-view image feature maps $\{\mathbf{F}_e^i\}_{i=1}^K$ and $\{\{\mathbf{F}_n^{j,i}\}_{i=1}^K\}_{j=1}^{N_n}$ are first extracted with a shared convolutional backbone, referred to as the Image Encoder. Then, the image features are transformed to BEV features $\mathbf{B}_e \in R^{H \times W \times C}$ and $\{\mathbf{B}_n^j\}_{j=1}^{N_n}$ with a PV2BEV module, which is implemented as an LSS operation Philion & Fidler (2020) following MapTRv2. Note that for each non-ego BEV feature $\mathbf{B}_n^j$, a coordinate transformation matrix derived from the camera parameters is applied to it to align with the ego BEV feature $\mathbf{B}_e$.

**Uncertainty-guided Multi-Vehicle Information Integrator.** The Uncertainty-guided Multi-Vehicle Information Integrator (UMVII) employs a progressive two-stage approach to process complex multi-vehicle data. In the **first stage**, each non-ego feature $\mathbf{B}_n^j$ is fused individually with the ego feature $\mathbf{B}_e$ via a Cross Fusion module to generate pairwise fused intermediate features $\mathbf{B}_f^j$. This provides a preliminary yet comprehensive integration of multi-vehicle information. To be specific, $\mathbf{B}_e$ and $\mathbf{B}_n^j$ are flattened to form ego and non-ego queries $\mathbf{Q}_e$ and $\mathbf{Q}_n^j \in \mathbb{R}^{HW \times C}$, and then fused

using deformable cross-attention, which can be briefly formulated as follows:

$$\mathbf{Q}_f^j = \mathcal{F}(\mathcal{F}(\mathbf{Q}_e, \mathbf{Q}_n^j), \mathbf{Q}_n^j), \tag{1}$$

where $\mathbf{Q}_f^j$ is the fused intermediate query feature. The fusion function $\mathcal{F}$ in Eq. 1 with two query features (denoted as $\mathbf{Q}_1$ and $\mathbf{Q}_2$) as input and a query feature (denoted as $\mathbf{Q}_3$) as output is defined as:

$$\mathbf{Q}_3 = \mathbf{Q}_1 + \sum_{i=1}^{N_{off}} \mathbf{W} \cdot \mathrm{DA}(\mathbf{Q}_2, \mathbf{R} + \mathbf{O}^i, \mathbf{B}_2), \tag{2}$$

$$\mathbf{O} = \mathrm{Offset\_Embedding}([\mathbf{Q}_1, \mathbf{Q}_2]), \mathbf{W} = \mathrm{Weight\_Embedding}([\mathbf{Q}_1, \mathbf{Q}_2]), \tag{3}$$

where $\mathrm{Offset\_Embedding}(x)$ and $\mathrm{Weight\_Embedding}(x)$ denote the convolutional embedding layers for sampling offsets and weights calculation, respectively. $[x_1, y_1]$ denotes the feature concatenation operation, while $\mathrm{DA}(\mathbf{Q}, x, \mathbf{B})$ is a deformable attention operation that utilize $\mathbf{Q}$ as a query to collect features at location $x$ on a BEV feature $\mathbf{B}$. $\mathbf{O}$ and $\mathbf{W}$ respectively represents the sampling offsets and weights. $N_{off}$ is the number of sampling offsets for each query. $\mathbf{R}$ denotes the reference points. $\mathbf{B}_2 \in \mathbb{R}^{H \times W \times C}$ is the BEV feature reshaped from $\mathbf{Q}_1 \in \mathbb{R}^{HW \times C}$. At this stage, the non-ego features separately provide complementary information to fill the gaps in the ego vehicle's perspective. However, additional information provided by these features may contain redundancies and is not optimally fused. Therefore, in the **second stage**, the ego feature and all these fused intermediate features are further fused under the guidance of the segmentation uncertainty to obtain the final BEV feature $\mathbf{B}_f$. Note that to better align with the fused intermediate features, the ego feature is first self-enhanced via deformable self-attention:

$$\hat{\mathbf{Q}}_e^j = \mathcal{F}(\mathcal{F}(\mathbf{Q}_e, \mathbf{Q}_e), \mathbf{Q}_e), \tag{4}$$

where $\hat{\mathbf{Q}}_e \in \mathbb{R}^{HW \times C}$ is the enhanced ego query feature, which is reshaped back to form the enhanced ego BEV feature $\hat{\mathbf{B}}_e \in \mathbb{R}^{H \times W \times C}$. Then, for each BEV feature to fuse, a segmentation uncertainty-based weight map is estimated, representing the importance of each BEV pixel regarding the map structure to construct. Taking $\hat{\mathbf{B}}_e$ as an example, it is first processed by a convolution-based instance segmentation block to predict a set of instance masks $\mathbf{M}_e \in \mathbb{R}^{H \times W \times N_{ins}}$. Subsequently, the corresponding uncertainty-based weight map $\mathbf{U}_e \in \mathbb{R}^{H \times W \times 1}$ can be obtained following:

$$\mathbf{U}_e = \mathrm{Softmax}(CScore_e \cdot \mathbf{M}_e), \tag{5}$$

$$CScore_e = \mathrm{IS}(\frac{1}{H \times W} \sum_{i=0}^{H} \sum_{j=0}^{W} (\mathbf{M}_e \odot \mathbf{B}_e)_{i,j}), \tag{6}$$

where $\mathrm{IS}(x)$ denotes the Instance Scorer consists of several MLP layers. $CScore_e \in \mathbb{R}^{N_{ins} \times 1}$ are the estimated instance confidence scores. $\mathrm{Softmax}(x)$ is the channel-wise softmax operation. $\odot$ stands for the outer product operation. Following similar paradigms, the non-ego fusion weight maps $\{\mathbf{U}_f^j\}_{j=1}^{N_n}$ can be obtained. The final fused BEV feature $\mathbf{B}_f$ can be obtained following:

$$\mathbf{B}_f = \mathbf{U}_e \cdot \hat{\mathbf{B}}_e + \sum_{j=1}^{N_n} \mathbf{U}_f^j \cdot \mathbf{B}_f^j. \tag{7}$$

**Segmentation Prior Guided Map Decoder.** Once the fused BEV feature $\mathbf{B}_f$ is obtained through UMVII, it is processed by our proposed Segmentation Prior Guided Map Decoder (SPMDec) to generate the vectorized map. SPMDec incorporates a key mechanism, namely, the Segmentation-guided Query Initialization (SQI), which is designed to function as a flexible add-on to existing map decoders. In our experiments, we adopt the map decoder architecture from MapTRv2 Liao et al. (2023), consisting of map queries and multiple decoder layers. For the design of map queries, a hierarchical query embedding scheme is utilized to explicitly encode each map element, which defines a set of instance-level queries $\{q_i^{ins}\}_{i=1}^{N_{ins}}$ and a set of point-level queries $\{q_j^{pt}\}_{j=1}^{N_{pt}}$ shared by all instances. The hierarchical query of $j$-th point of $i$-th map element $q_{ij}^{hie}$ is formulated as:

$$q_{ij}^{hie} = q_i^{ins} + q_j^{pt}, \tag{8}$$

---

**Algorithm 1:** Uncertainty-aware Non-ego Vehicle Selection Strategy

---

**Input:** Pixel-wise map element prediction probability $\mathbf{P}$; radius $d$; threshold $\tau$.

**Output:** Non-ego Vehicle IDs.$\{\mathcal{I}_n^j\}_{j=1}^{N_n}$.

1 **Uncertain Area Selection step:**
2 Form each $\mathbf{\Omega}_{i,j}$ for each BEV pixel based on $d$ and $\tau$;
3 Compute $\mathbf{U}$ following Eq. 11;
4 $\mathbf{U}_{grid} = \text{AvgPooling}(\mathbf{U})$;
5 Select grids with top-$N_n$ grid-wise uncertainty as uncertain areas;
6 **Uncertainty-aware Non-ego Vehicle step:**
7 Take center coordinates of uncertain areas as sample points;
8 Select ids of the nearest non-ego vehicles to every sample point;

---

Differently, in the proposed SPMDec, a Segmentation-guided Query Initialization (SQI) mechanism is designed to inject multi-vehicle prior information into the map queries. To be specific, the fused BEV feature $\mathbf{B}_f \in \mathbb{R}^{H \times W \times C}$ is processed by a convolutional block to predict a set of instance masks $\mathbf{M}_f \in \mathbb{R}^{H \times W \times N_{ins}}$. The set of multi-vehicle prior information injected instance queries $\{q_i^{mv,ins}\}_{i=1}^{N_{ins}}$ are then calculated as follows:

$$\{q_i^{mv,ins}\}_{i=1}^{N_{ins}} = \text{AvgPooling}(\mathbf{M}_f \odot \mathbf{B}_f), \tag{9}$$

where $\text{AvgPooling}(x)$ denotes the spatial average pooling operation. Thus the multi-vehicle prior information injected hierarchical query of $j$-th point of $i$-th map element $q_{ij}^{hie}$ is formulated as:

$$q_{ij}^{mv,hie} = q_i^{mv,ins} + q_j^{pt}, \tag{10}$$

Then each decoder layer uses self-attention and cross-attention to update the map queries. Finally, a simple prediction head consisting of a classification branch and a point regression branch is adopted to output instance classification scores and normalized BEV coordinates for each map instance, respectively.

### 3.4 UNCERTAINTY-AWARE NON-EGO VEHICLE SELECTION STRATEGY

In addition to UMVMap-Net which manages to give a superior map construction by properly integrating multi-vehicle information, we further proposed an Uncertainty-aware Non-ego Vehicle Selection (UNVS) Strategy to improve performance by selecting non-ego vehicles that provide complementary information in regions where the ego vehicle's perception is highly uncertain. The overall process of the UNVS Strategy is illustrated in Fig. 4, which is mainly composed of two steps, namely, an Uncertain Area Selection step and an Uncertainty-aware Non-ego Vehicle step. During the **Uncertain Area Selection step**, we split the BEV space into $N_h \times N_w$ grids and estimate the uncertainty for each grid. To be specific, a pre-trained MapTRv2 Liao et al. (2023) model is first employed to obtain the pixel-wise map element prediction probability $\mathbf{P} \in \mathbb{R}^{H \times W \times 1}$ for the ego BEV feature. To obtain the grid-wise uncertainty, a pixel-wise uncertainty map is first estimated. For each pixel in the $i$-th row and the $j$-th column, the pixel-wise map element prediction probability values greater than a threshold $\tau$ within a radius $d$ of this pixel are gathered into an adjacent set $\mathbf{\Omega}_{i,j}$. The uncertainty $\mathbf{U}_{i,j}$ of this pixel is calculated following:

$$\mathbf{U}_{i,j} = \frac{1}{|\mathbf{\Omega}_{i,j}|} \sum_{p \in \mathbf{\Omega}_{i,j}} (1 - p), \tag{11}$$

where $|\mathbf{\Omega}_{i,j}|$ denotes the size of the set. Then, the uncertainty values of pixels within corresponding grids are averaged to obtain the grid-wise uncertainty map

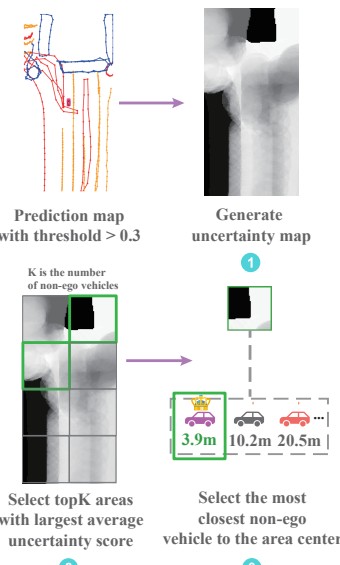

Figure 4: Uncertainty-aware Non-ego Vehicle Selection Strategy.

$\mathbf{U}_{grid} \in \mathbb{R}^{N_h \times N_w}$. With the aim of picking $N_n$ non-ego vehicles, the $N_n$ areas with the highest uncertainty values are selected as the uncertain areas. After that, in the **Uncertainty-aware Non-ego Vehicle step**, the center coordinates of the selected uncertain areas are taken as the sample points. For each of the $N_n$ sample points, the nearest non-ego vehicle is selected. A brief algorithm overview is given in Alg.1.

### 3.5 OPTIMIZATION

To optimize the proposed UMVMap, we follow the optimization setting of MapTRv2 Liao et al. (2023), which utilizes a basic one-to-one set prediction loss and two auxiliary losses including a one-to-many set prediction loss and a dense prediction loss. To be specific, the basic one-to-one set prediction loss is composed of three parts, including a classification loss, a point-to-point loss and an edge direction loss:

$$\mathcal{L}_{one2one} = \lambda_c \mathcal{L}_{cls} + \lambda_p \mathcal{L}_{p2p} + \lambda_d \mathcal{L}_{dir}, \tag{12}$$

where $\lambda_c$, $\lambda_p$ and $\lambda_d$ are the weights for balancing different loss terms. The classification loss is implemented as a Focal Loss. The point-to-point loss is defined as the Manhattan distance computed between each assigned point pair after hierarchical bipartite matching. The edge direction loss is designed as the cosine similarity of each pair of predicted edge and ground-truth edge. The one-to-many set prediction loss $\mathcal{L}_{one2many}$ is also adopted. Besides, dense prediction loss $\mathcal{L}_{dense}$ is utilized to further leverage semantic and geometric information, which consists of a depth prediction loss, a BEV segmentation loss and a PV segmentation loss. Note that $\mathcal{L}_{dense}$ is applied on $\hat{\mathbf{B}}_e$, $\{\mathbf{B}_f^j\}_{j=1}^{N_n}$ and $\mathbf{B}_f$. In addition, a confidence score loss $\mathcal{L}_{cscore}$ calculated between the estimated $CSore$ and the corresponding ground truth value is proposed. Finally, the overall loss for optimizing UMVMap can be defined as follows:

$$\mathcal{L} = \beta_o \mathcal{L}_{one2one} + \beta_m \mathcal{L}_{one2many} + \beta_d \mathcal{L}_{dense} + \beta_c \mathcal{L}_{cscore}. \tag{13}$$

## 4 EXPERIMENTS

### 4.1 EXPERIMENTAL SETTINGS

**Datasets.** We evaluate our proposed method using the popular and large-scale nuScenes Caesar et al. (2020) dataset. The nuScenes dataset includes 2D city-level global vectorized maps and 1,000 scenes, each approximately 20 seconds long. Key samples are annotated at 2Hz, with each sample providing RGB images from six cameras, covering a 360° horizontal field of view around the ego-vehicle. The dataset is split into 28K frames for training and 6K frames for validation. For all of the experiments on nuScenes, we utilize the whole validation set as Full Validation Set (with 6019 samples), while utilizing the validation samples with available non-ego vehicles as Partial Validation Sets (with 2667 samples).

**Metrics.** We conduct evaluation with Chamfer distance based Average Precision (AP) following previous mainstream works Li et al. (2022); Liao et al. (2023) for fair comparisons. The AP is calculated under the average of three Chamfer distance thresholds of 0.5, 1.0, and 1.5 meters. The perception ranges are [-15.0m, 15.0m] for the X-axis and [-30.0m, 30.0m] for the Y-axis. Three types of map instances are selected for HD map construction, including pedestrian crossing, lane divider, and road boundary. Besides, the mean Average Recall (mAR) is employed to evaluate the performance of non-ego vehicle selection methods.

**Implementation Details.** We employ ResNet50 He et al. (2016) as the backbone network for image processing. For the nuScenes dataset, images with dimensions of $1600 \times 900$ are resized by a factor of 0.5. The default settings for instance queries, point queries, and decoder layers are 50, 20, and 6, respectively. We use the AdamW optimizer with a learning rate of $6 \times 10^{-4}$ and a weight decay of 0.01. All models are trained using 8 80GB NVIDIA Tesla A100 GPUs, with a batch size of 4 per node, leading to a total batch size of 32. Following Liao et al. (2023), we set $\lambda_c = 2$, $\lambda_p = 5$, $\lambda_d = 0.05$. For the overall loss, we set $\beta_o = 1$, $\beta_m = 1$, $\beta_d = 1$, $\beta_c = 1$.

Table 1: Comparison with SOTA on nuScenes. The Full Validation Sets with 6019 samples denotes the whole validation set of nuScenes, while the Partial Validation Sets with 2667 samples is the validation samples with available non-ego vehicles.

| - | - | Full Validation Set | | | | Partial Validation Set | | | |
|---|---|---|---|---|---|---|---|---|---|
| Methods | Backbone | $AP_{div}(\uparrow)$ | $AP_{ped}(\uparrow)$ | $AP_{bnd}(\uparrow)$ | $mAP(\uparrow)$ | $AP_{div}(\uparrow)$ | $AP_{ped}(\uparrow)$ | $AP_{bnd}(\uparrow)$ | $mAP(\uparrow)$ |
| HDMapNet Li et al. (2022) | EB0 | 14.4 | 21.7 | 33.0 | 23.0 | - | - | - | - |
| MapTR Liao et al. (2022) | R50 | 46.3 | 51.5 | 53.1 | 50.3 | - | - | - | - |
| MapVR Zhang et al. (2024a) | R50 | 56.2 | 56.5 | 60.1 | 57.6 | - | - | - | - |
| PivotNet Ding et al. (2023) | R50 | 47.7 | 54.4 | 51.4 | 51.2 | - | - | - | - |
| BeMapNet Qiao et al. (2023) | R50 | 62.3 | 57.7 | 59.4 | 59.8 | - | - | - | - |
| MGMap Liu et al. (2024) | R50 | 61.8 | 65.0 | 67.5 | 64.8 | - | - | - | - |
| MapTRv2 Liao et al. (2023) | R50 | 59.8 | 62.4 | 62.4 | 61.5 | 58.5 | 63.1 | 62.1 | 61.2 |
| UMVMap (ours) | R50 | **70.5** | **69.4** | **71.8** | **70.6** | **69.6** | **73.8** | **72.9** | **72.1** |

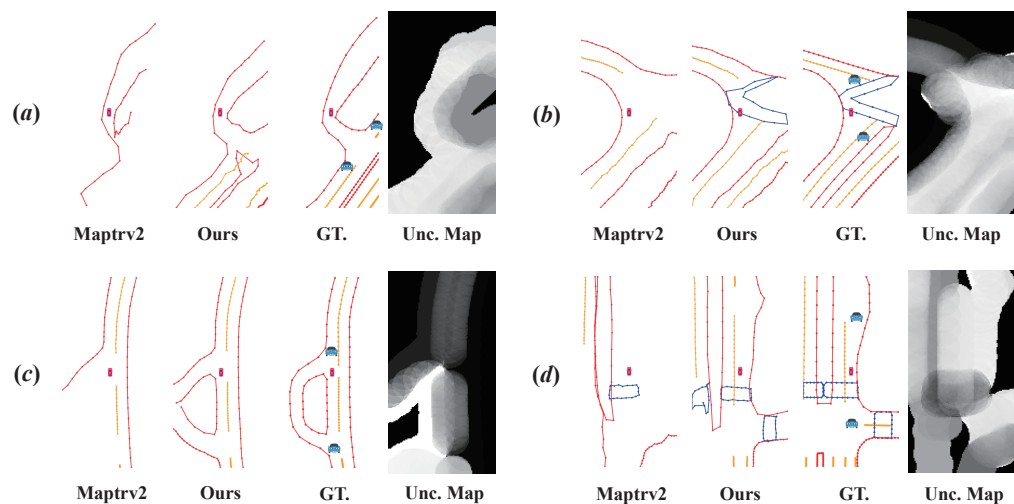

Figure 5: Comparison of qualitative results on the nuScenes dataset. "Unc. Map" means uncertainty map. The blue car logos in GT. figure show the locations of non-ego vehicles.

## 4.2 MAIN RESULTS

As reported in Table 1, we compare the proposed UMVMap with state-of-the-art (SOTA) methods on the nuScenes validation set. The results demonstrate that UMVMap outperforms previous methods and achieves the best overall performance. Specifically, compared to the baseline MapTRv2 Liao et al. (2023), UMVMap shows an improvement of 9.1% and 9.9% mAP respectively on the full and partial validation sets by integrating complementary information from multiple vehicles. Additionally, UMVMap surpasses the current leading approach, MGMap Choi et al. (2024), with a further gain of 5.8% mAP.

## 4.3 ABLATION STUDIES AND HYPER-PARAMETER ANALYSIS

We conducted ablation studies to evaluate the contributions of the core components of UMVMap. The training was conducted on the full nuScenes training set for 24 epochs. Evaluation was performed on both the Full Validation Set and Partial Validation Set.

**Contribution of Main Components.** Table 2 shows the impact of key components in UMVMap-Net, including the two-stage Uncertainty-guided Multi-Vehicle Information Integrator (UMVII) and

Table 2: Ablation analysis of key components on nuScenes.

| Methods | | | Full Validation Set | | | |
|---|---|---|---|---|---|---|
| UMVII(1) | UMVII(2) | SQI | $AP_{div}(\uparrow)$ | $AP_{ped}(\uparrow)$ | $AP_{bnd}(\uparrow)$ | $mAP(\uparrow)$ |
| × | × | × | 41.1 | 40.2 | 44.8 | 42.0 |
| ✓ | × | × | 45.0 | 44.0 | 48.8 | 45.9 |
| ✓ | × | ✓ | 45.1 | 47.1 | 54.2 | 48.8 |
| ✓ | ✓ | ✓ | 50.6 | 48.7 | 56.9 | 52.1 |

Table 3: Comparison of non-ego vehicle selection method.

| - | Full Validation Set | | | | | Partial Validation Set | | | | |
|---|---|---|---|---|---|---|---|---|---|---|
| Methods | $AP_{div}(\uparrow)$ | $AP_{ped}(\uparrow)$ | $AP_{bnd}(\uparrow)$ | $mAP(\uparrow)$ | $mAR(\uparrow)$ | $AP_{div}(\uparrow)$ | $AP_{ped}(\uparrow)$ | $AP_{bnd}(\uparrow)$ | $mAP(\uparrow)$ | $mAR(\uparrow)$ |
| Random | 65.8 | 63.3 | 66.1 | 65.1 | 75.2 | 66.2 | 70.1 | 69.2 | 68.5 | 79.6 |
| Closest | 68.3 | 65.6 | 68.5 | 67.6 | 77.2 | 68.7 | 70.8 | 71.2 | 70.2 | 80.2 |
| UNVS | **70.5** | **69.4** | **71.8** | **70.6** | **78.2** | **69.6** | **73.8** | **72.9** | **72.1** | **80.9** |

the Segmentation-guided Query Initialization (SQI) mechanism. The evaluation process involves progressively adding each component to a baseline model. The first row serves as a baseline, which employs a naive approach that directly concatenates and fuses the ego and non-ego features with several MLPs. Incorporating the first stage of UMVII into the baseline yields performance gains of 3.9% mAP. This demonstrates that the interaction between ego and non-ego features enhances the accuracy of vectorized map construction. The inclusion of SQI brings further improvements of 2.9% mAP, showcasing the benefits of introducing segmentation prior knowledge into the instance queries during the map decoding process. Additionally, including the second stage of UMVII which leverages semantic uncertainty to guide multi-vehicle information integration brings further improvements of 3.3% mAP.

**Effect of Non-ego Vehicle Selection Method.**   As shown in Table 3, we investigate the effectiveness of the non-ego vehicle selection methods by comparing the proposed Uncertainty-aware Non-ego Vehicle Selection (UNVS) strategy with several of its variants, which involves: (i) Random: selecting $N_n$ non-ego vehicles with random distances from the ego vehicle. (ii) Closest: selecting the $N_n$ non-ego vehicles closest to the ego vehicle. For each pivot point, the nearest non-ego vehicle is selected. It can be observed that by selecting non-ego vehicles with a clearer information complementing purpose under the guidance of uncertainty, our proposed UNVS strategy yields a more significant performance improvement.

**Hyper-parameter Analysis on Non-ego Vehicle Number.**   A hyper-parameter analysis is conducted on the non-ego vehicle number $N_n$. As can be seen from Table 4, increasing $N_n$ from 1 to 2 results in a notable performance improvement of 4.7% and 5.6% mAP on the full and partial validation sets, respectively. Further increasing $N_n$ from 2 to 3 does not yield any performance gain. This suggests that while incorporating additional non-ego vehicles provides valuable information, the information gain approaches an upper limit especially when we adopt a proper vehicle selection strategy. Information from two non-ego vehicles selected by our UNVS strategy can be sufficient.

**Hyper-parameter Analysis on Non-ego Vehicle Selection Time Window.**   We further investigate the impact of the choice of the non-ego vehicle selection time window in Table 5. We aim to introduce more variety to the street scenes involving non-ego vehicles, such as different times of day with reduced traffic noise. To achieve this, we excluded non-ego vehicles from scenes recorded within the same 30-minute window. The results show that this yields the best performance (30~$\infty$ time window), as seen by the improvement in all metrics. On the other hand, limiting the model to

Table 4: Hyper-parameter analysis on non-ego vehicle number $N_n$.

| - | Full Validation Set | | | | Partial Validation Set | | | |
|---|---|---|---|---|---|---|---|---|
| $N_n$ | $AP_{div}(\uparrow)$ | $AP_{ped}(\uparrow)$ | $AP_{bnd}(\uparrow)$ | $mAP(\uparrow)$ | $AP_{div}(\uparrow)$ | $AP_{ped}(\uparrow)$ | $AP_{bnd}(\uparrow)$ | $mAP(\uparrow)$ |
| 1 | 66.7 | 64.0 | 67.1 | 65.9 | 64.6 | 66.8 | 68.0 | 66.5 |
| 2 | 70.5 | 69.4 | 71.8 | 70.6 | 69.6 | 73.8 | 72.9 | 72.1 |
| 3 | 69.7 | 70.4 | 71.7 | 70.6 | 68.1 | 74.3 | 72.8 | 71.7 |

Table 5: Hyper-parameter analysis on time window (minute) for non-ego vehicle selection.

| Time Window | $AP_{div}(\uparrow)$ | $AP_{ped}(\uparrow)$ | $AP_{bnd}(\uparrow)$ | $mAP(\uparrow)$ |
|---|---|---|---|---|
| $0\sim\infty$ | 69.8 | 67.7 | 70.5 | 69.3 |
| $30\sim\infty$ | 70.5 | 69.4 | 71.8 | 70.6 |
| $0\sim30$ | 68.5 | 66.7 | 68.7 | 68.0 |

the first 30 minutes ($0\sim30$) leads to a significant performance drop across all metrics. This infers that most of the valuable non-ego vehicles are outside the $0\sim30$ time window.

### 4.4 QUALITATIVE ANALYSIS

Fig. 5 illustrates the comparison of the qualitative results generated by the MapTRv2 baseline and our proposed UMVMap. The uncertainty map for selecting non-ego vehicles is also illustrated in each example. It can be observed that by properly selecting and then integrating multi-vehicle information, UMVMap captures finer and more accurate details, which proves the effectiveness of our proposed method.

## 5 CONCLUSION

In this study, we present UMVMap, an Uncertainty-aware Multi-Vehicle Vectorized Map Construction framework designed to effectively mitigate uncertainties in vectorized map construction by leveraging relevant non-ego vehicle information. UMVMap is composed of two key components: the Uncertainty-aware Multi-Vehicle Vectorized Map Construction Network (UMVMap-Net), which optimally integrates data from multiple vehicles, and the Uncertainty-aware Non-ego Vehicle Selection (UNVS) strategy, which identifies and incorporates the most informative non-ego data to minimize uncertainty. Comprehensive evaluations conducted on the nuScenes dataset demonstrate that UMVMap significantly outperforms the single-vehicle MapTRv2 baseline by a margin of 9.1% and 9.9% respectively on the full and partial validation sets, with its components showing notable effectiveness and robustness. Looking ahead, vehicle-to-vehicle collaborative perception for vectorized map construction remains a promising direction for future exploration, with substantial potential for further advancements in autonomous driving.

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
