# OpenReview forum: "UMVMap: Improving Vectorized Map Construction via Multi-vehicle Perspectives"
_ICLR.cc/2025/Conference — Submitted to ICLR 2025_

### Official Review · Reviewer_oXzp · 2024-10-28

**Soundness:** 3
**Presentation:** 3
**Contribution:** 2
**Rating:** 5
**Confidence:** 2

**Summary:**

The paper presents a novel framework, Uncertainty-aware Multi-Vehicle Vectorized Map Construction (UMVMap), aimed at enhancing vectorized map construction for autonomous driving by integrating data from multiple vehicles. The authors highlight the limitations of existing DETR-based methods that rely solely on data from a single ego vehicle, leading to perceptual uncertainty in complex environments. UMVMap addresses these issues by employing two main components: the Uncertainty-aware Multi-Vehicle Vectorized Map Construction Network (UMVMap-Net) and the Uncertainty-aware Non-ego Vehicle Selection (UNVS) strategy. The proposed framework shows significant performance improvements over the single-vehicle baseline (MapTRv2), with experimental results indicating enhancements of 9.1% and 9.9% on the full and partial validation sets of the nuScenes dataset.

**Strengths:**

•	Novel Framework: The introduction of UMVMap, which utilizes multi-vehicle data for vectorized map construction, is a significant contribution to the field. Emphasizing the importance of effectively aggregating information from diverse views.
•	UNVS Strategy: The development of the UNVS strategy allows for the identification and selection of non-ego vehicles that provide complementary information, particularly in areas with high uncertainty.
•	Comprehensive Evaluations: The thorough evaluation of UMVMap against a robust dataset establishes its superiority over existing single-vehicle methods and highlights the effectiveness of its components through ablation studies.

**Weaknesses:**

•	Limited Scope of Evaluation: While the framework performs well on the nuScenes dataset, additional evaluations on diverse datasets or real-world scenarios would strengthen the claims of generalizability.
•	Complexity of Implementation: The proposed framework may involve a more complex implementation compared to simpler methods, which could deter practical adoption.
•	Scalability Concerns: The paper does not adequately address the scalability of the system in scenarios with a significantly higher number of vehicles, which could impact performance and computational resources.
•	Reproduce concern: code and datasets were not provided in this paper, and making it unclear whether the experimental results can be reproduced.

**Questions:**

•	Expanded Evaluation: Consider incorporating additional datasets to demonstrate the robustness and versatility of UMVMap in various environments. Show more information on the computational cost of the method, particularly in larger datasets or more complex domains.
•	Simplification of Framework: Explore opportunities to simplify the implementation process or provide more detailed guidelines for practical deployment.
•	Address Scalability: Discuss potential strategies or modifications to handle scalability issues as the number of vehicles increases in real-world applications.
•	Ensure reproducibility: Access to project dataset source and code is expected to make sure results are reproducable.

---

> ### Comment · Reviewer_oXzp · 2024-11-26
> **To authors**
>
> After read the rebuttal and other reviewers, I would like to keep my score.

---

### Official Review · Reviewer_VrQt · 2024-11-02

**Soundness:** 3
**Presentation:** 2
**Contribution:** 2
**Rating:** 5
**Confidence:** 4

**Summary:**

This paper proposed a novel framework to extend the previous single-agent map construction framework into a multi-agent scenario where they could leverage other cameras from the collaborative vehicles to handle the long-range and occlusion issues limited by the single-agent viewpoint. They proposed a uncertainty-aware method to fuse the multi-vehicle information.

**Strengths:**

The paper is novel in the field of multi-agent collaborative map construction task since there is no much literature discussing on map construction with the help of multi-agent collaboration. The paper is overall well-written and easy to follow, and the enhancement is significant as described by the authors.

**Weaknesses:**

Overall, the contribution is limited at the aspect that, it is not surprising to know if we leverage multiple cameras at different view points the results will improve. In fact, the paper seems focusing less on the problem of multi-agent collaboration, i.e., the communication volume, localization error, perturbations that are essential to the collaboration of different agents. We could regard that the problem becomes easier with the help of multi-agent, but the paper does not address much on the issues inherent of multi-agent scenario, which limits the scope of the contribution.

**Questions:**

Some specific questions:
1. The paper lacks of generalizability on the proposed methods (only show in one baseline and one dataset), and it feels necessary to augment their experiments by leveraging the recent advancement of cooperative datasets (i.e., DAIR-V2X) and apply for more baselines.
2. The paper lacks of discussion on why having more than 2 agents result in performance drop, which is counter-intuitive. This could be a place where the method could be further enhanced (only selecting the closest vehicle, which is probably vanilla).
3. As mentioned above, the paper should have some more discussions on the viability when applying the framework to real-world online inference scenario (where it's the core of multi-agent problem, what/where/how to communicate with each other).

---

### Official Review · Reviewer_ykxK · 2024-11-04

**Soundness:** 2
**Presentation:** 2
**Contribution:** 2
**Rating:** 5
**Confidence:** 3

**Summary:**

This paper explores a new task: multi-vehicle cooperative vectorized map construction, aiming to leverage information from non-ego vehicles to mitigate perceptual uncertainty in complex scenarios. The paper introduces a framework called Uncertainty-aware Multi-vehicle Vectorized Map Construction (UMVMap) for this task. UMVMap comprises two components: the Uncertainty-aware Multi-vehicle Vectorized Map Construction Network (UMVMap-Net) and the Uncertainty-aware Non-ego Vehicle Selection (UNVS) strategy to optimally leverage information from multiple vehicles. Experimental results demonstrate that UMVMap outperforms state-of-the-art single-vehicle methods.

**Strengths:**

1. This paper presents UMVMap, a pioneering work exploring vehicle-to-vehicle collaborative perception in vectorized map construction. Experimental results demonstrate that UMVMap outperforms state-of-the-art single-vehicle methods.

2. UMVMap consists of several key components: the Uncertainty-guided Multi-Vehicle Information Integrator (UMVII), the Segmentation-guided Query Initialization Mechanism (SQI), and the Uncertainty-aware Non-ego Vehicle Selection Strategy (UNVS). Ablations demonstrate the effectiveness of each component.

**Weaknesses:**

1. The proposed UMVMap lacks significant novelty. It incorporates the idea of confidence-aware communication from existing collaborative perception methods, such as Where2comm [1], to integrate intermediate features from the single-vehicle method MapTR [2] for the cooperative task. Please answer the following questions in the rebuttal with details: (1) What are the challenges of the cooperative vectorized map construction task? (2) What are the key contributions of the proposed UMVMap compared to existing collaborative perception and map construction methods? (3) How have these contributions addressed the aforementioned challenges?

2. Limited discussion is provided on real-world implementation. Communication efficiency is essential in the cooperative autonomous driving community. In this work, raw sensor data from non-ego vehicles is transmitted to the ego vehicle, leading to significant communication costs and transmission delays.

3. There are several errors in the paper, such as the incorrect citation of StreamMapNet and duplicate citations of MapVR.

[1] Hu et al. Where2comm: Communication-Efficient Collaborative Perception via Spatial Confidence Maps. NeurIPS 2022. \
[2] Liao et al. MapTR: Structured Modeling and Learning for Online Vectorized HD Map Construction. ICLR 2023.

**Questions:**

1. Please address the weaknesses.

2. What is the communication volume of the proposed UMVMap, and how does the method handle communication delays in real-time applications?

---

### Official Review · Reviewer_EgVD · 2024-11-04

**Soundness:** 2
**Presentation:** 2
**Contribution:** 2
**Rating:** 3
**Confidence:** 4

**Summary:**

In this article, the authors present a novel online mapping framework called Uncertainty-aware Multi-Vehicle Vectorized Map Construction (UMVMap). This framework addresses the uncertainty in ego-vehicle HD map construction by integrating information from surrounding traffic participants.

According to the description provided in the paper, UMVMap comprises two components:

An uncertainty-aware multi-vehicle vectorized map construction network (UMVMap-Net) consists of two sub-modules: the Uncertainty-guided Multi-Vehicle Information Integrator (UMVII), which integrates data from multiple vehicles, and the Segmentation Prior Guided Map Decoder (SPMDec), which decodes the integrated information. As noted by the authors in the paper, this structure enables UMVMap-Net to select data from various vehicles using an optimal strategy.

In addition, they utilize a Segmentation Prior Guided Map Decoder (UNVS) strategy to aid the model in identifying other traffic participants in regions where the ego-vehicle's perception results are highly uncertain.

Their comprehensive evaluation on the nuScenes dataset indicates that UMVMap has achieved what appears to be acceptable performance.

**Strengths:**

The authors propose a Vehicle-to-Vehicle (V2V) perception paradigm for the task of online mapping.

**Weaknesses:**

From my perspective, there are several concerns regarding this article.

Firstly, I believe that the Vehicle-to-Vehicle (V2V) mapping approach lacks practical value. The deployment costs associated with this type of perception are prohibitively high, and it requires stable communication. Furthermore, this method would introduce significant latency, a critical issue that the authors have not addressed anywhere in the paper.

Secondly, regarding the dataset selected by the authors, to the best of my knowledge, the nuScenes dataset is inadequate for supporting vehicle-to-vehicle (V2V) online mapping tasks. Additionally, the authors have not included specific details in the dataset section of the paper about how they processed the nuScenes dataset, which is typically utilized for single-vehicle perception tasks, to adapt it for multi-vehicle perception tasks.

In addition, in the StreamMapNet work, the authors implemented a more rational division of the online mapping dataset. However, it seems that they are still utilizing the previous dataset division, which may result in unfairness in the evaluation of the results.

Moreover, the authors' experiments lack comprehensive comparisons with the latest works, such as HIMap, MapTracker, and HRMapNet.

Lastly, there are several unclear points and errors in the authors' writing. For instance, in Figure 2, I cannot understand how the number of ego vehicles can exceed 10,000.

**Questions:**

1. Please explain in detail how the authors used the nuScenes dataset to complete V2V online mapping.

2. Why didn't the authors conduct more experiments on Argoverse2?

---

### Meta-Review · Area_Chair_jKWm · 2024-12-16

**Metareview:**

This work proposes a new method for map construction for autonomous vehicles. It introduces an uncertainty-aware strategy for vehicle selection during mapping.
The verdict on novelty is split. While the reviewers tend to see the work as novel, they also point out the limitations of practical usefulness and the lack of novelty in the underlying design choices. The reviewers also point out further weaknesses, such as the inadequacy of the selected dataset, practical limitations, or lack of comparisons with recent work in the space.

The concerns raised by the reviewers seem largely legitimate, and the authors have chosen not to participate in the reviewer discussion, thus not addressing the reviewer's concerns.

**Additional Comments On Reviewer Discussion:**

The authors did not participate in the discussion, and no new revisions have been submitted.

---

### Decision · Program_Chairs · 2025-01-22

Reject